# Advances in Animal Models for Studying Bone Fracture Healing

**DOI:** 10.3390/bioengineering10020201

**Published:** 2023-02-03

**Authors:** Hui Gao, Jinming Huang, Quan Wei, Chengqi He

**Affiliations:** 1Rehabilitation Medicine Center and Institute of Rehabilitation Medicine, West China Hospital, Sichuan University, Chengdu 610041, China; 2Key Laboratory of Rehabilitation Medicine in Sichuan Province, Chengdu 610041, China

**Keywords:** fracture healing, osteoporosis, animal models

## Abstract

Fracture is a common traumatic injury that is mostly caused by traffic accidents, falls, and falls from height. Fracture healing is a long-term and complex process, and the mode of repair and rate of healing are influenced by a variety of factors. The prevention, treatment, and rehabilitation of fractures are issues that urgently need to be addressed. The preparation of the right animal model can accurately simulate the occurrence of fractures, identify and observe normal and abnormal healing processes, study disease mechanisms, and optimize and develop specific treatment methods. We summarize the current status of fracture healing research, the characteristics of different animal models and the modeling methods for different fracture types, analyze their advantages and disadvantages, and provide a reference basis for basic experimental fracture modeling.

## 1. Introduction

Fractures are a common and serious traumatic injury. Fracture healing is a long-term and sophisticated process, and its osteogenesis and healing time are influenced by a variety of factors (such as blood supply, stability, and inflammation), with 5–10% of fractures failing to heal [1,2]. The main sites of fracture are the hip and the spine, with the former being the most serious. Because most hip fractures are fragility fractures, they tend to occur in elderly people with osteoporosis and are one of the main causes of disability among them. It was found that 22% of women and 33% of men died in the first year after hip fracture [3]. According to epidemiological data, more than 1.6 million patients worldwide suffer hip fractures annually [4]. In 2019, more than 67,000 hip fractures occurred in England, Wales, and Northern Ireland, costing the National Health Service (NHS) hospitals approximately 1.1 billion GBP a year [5]. In addition, the annual incidence of spinal fractures was 29.3/1000 for women and 13.6/1000 for men aged 75–79 years [6]. These patients frequently suffer from pain [7,8], spinal deformities [9], reduced pulmonary function [8,10], and restricted activities of daily living [11], all of which severely impact their quality of life. Overall, fractures represent a major cause of disability and a heavy burden on global health care. Therefore, it is of great importance to study how to accelerate fracture healing, reduce fracture complications, and decrease the disability rate.

Currently, with the use of omics increasing and multidisciplinary integration deepening, there is a shift in fracture research toward gene therapy and molecular therapy; the development of novel materials, adjuvants, and biologicals; and the exploration of intelligent drug delivery systems [12,13,14]. However, there is still a gap between clinical application and mechanistic studies, and the potential adverse effects and unestablished indications impede the optimization of clinical treatment. To bridge this gap and therefore improve overall patient prognosis, improved treatments must be translated into clinical practice. Preclinical fracture studies, i.e., animal-based fracture studies, are a necessary step.

The selection of suitable animal models can maximally simulate the occurrence of fractures and induce normal or abnormal healing processes, which facilitates the study of bone healing mechanisms (especially Haversian system-mediated biomechanical functional reconstruction), promotes the development of treatment methods, and ultimately guides clinical practice. Here, we summarize the current status of research on fracture healing, the characteristics of different animal models, the latest modeling methods and details for fracture models at different sites, and pathological fracture models (such as osteoporotic fractures and nonunion fractures). The aim of our study is to assist investigators in selecting the most appropriate animal model and to increase the validity and reliability of preclinical studies.

## 2. Search Strategy

We searched PubMed, Embase, and the Cochrane Library for articles and reviews published in English between 2000 and 2022; although, older references were also used as appropriate.

We used the keywords “fractures” in combination with “mice”, “mouse”, “murine”, “rats”, “dogs”, “canine”, “rabbits”, “cows”, “sheep”, “goat”, “ovine” and “caprine”, “pigs”, “monkeys”, “rodents”, and “primates” and limited the search to animal studies. In addition, conference abstracts and books were also manually searched, and references included in articles and reviews were screened.

## 3. Mechanism of Fracture Healing

### 3.1. Three Key Elements in Fracture Healing

The repair of fractures is a process of calcification of bone tissue, which contains three interdependent elements: cells, organic matrix, and inorganic substances [15].

The cells existing in bone tissue are predominantly osteoblasts, osteocytes, osteoclasts, and osteogenic precursor cells (mesenchymal osteoprogenitor cells) [16]. Osteoblasts are responsible for laying down collagen, after which the calcification process begins and induces the formation of a solid, stable, crystalline inorganic phase in the organic phase [17]. Osteoblasts are transformed into osteocytes after being embedded by the matrix they produce, which has a sensory effect on mechanical stress [18]. Osteoclasts are involved in bone resorption and play an important role in clearing excess bone matrix and bone remodeling [19]. Osteogenic precursor cells are able to differentiate directly into osteoblasts in response to specific growth factors, or into chondrocytes to participate in endochondral ossification [20] (details in Section 3.2).

The organic bone matrix is dominated by type I collagen fibers and contains small amounts of collagen type III and collagen type V [16]. They have structural properties that form scaffolds for inorganic components and control the alignment, assembly, integrity, and mechanical properties in bone tissue [21]. Non-collagenous components include proteoglycans, decorin, glycoproteins, osteopontin, osteocalcin, bone sialoprotein, byglican, osteonectin, thrombospondin, fibronectin, and phospholipids [15,16,22]. Although the percentage is small (approximately 10%), they are important in compact bone because they affect the rate of bone formation and collagen bundle spacing [15].

The inorganic bone matrix is composed mainly of crystalline mineral salts and calcium in the form of hydroxyapatite, including 85% tricalcium phosphate, 10% calcium carbonate, and 5% fluorinated derivatives such as calcium fluoride and magnesium fluoride [16,23], whose functions are related to tensile strain transfer and mechanical support.

### 3.2. The Process of Indirect Fracture Healing

Indirect fracture healing (also known as secondary healing) occurs when the fracture site fails to meet rigid anatomic reduction with no gap formation, and is the repair modality for most fractures [24]. There are three phases of indirect fracture healing: the inflammatory phase, proliferative phase, and remodeling phase [1]. In addition, two mechanisms of fracture repair occur during indirect fracture healing, namely, endochondral ossification and intramembranous ossification [25]. The crucial difference between endochondral ossification and intramembranous ossification is the presence or absence of the chondrogenic phase [25]. Moreover, intramembranous ossification occurs usually at the distal and proximal ends of the fracture, or bone defects (e.g., tibial or calvarial defects), while endochondral ossification usually occurs in long bone fractures (e.g., the middle femur fracture) [26,27].

During endochondral ossification, osteogenic precursor cells migrate to the wound site in response to elevated levels of growth factors and cytokines, where they proliferate and differentiate into chondrocytes and produce cartilage matrix, followed by calcification and formation of woven bone [20]. Woven bone is a temporary structure composed of irregularly arranged collagen fibers and randomly dispersed crystals [16]. In the remodeling phase, woven bone is gradually transformed into lamellar bone, restoring the mechanical integrity of the healing site [1] (Figure 1).

In contrast, during intramembranous ossification, cells in the inner periosteal osteogenic layer contribute to osteogenesis instead of cartilaginous tissue. Specifically, available skeletal stem cells and osteoprogenitor cells begin to proliferate during the early proliferative phase [28,29]. Subsequently, the adjacent osteoprogenitor cells form an ossification center and initiate the osteogenesis process [30] (Figure 2). Notably, the biomechanical strength of bone formed by intramembranous ossification is inferior to that of endochondral osteogenesis [16,30] (Figure 2).

Various factors affect fracture healing. Internal factors include the nature and degree of trauma, local soft tissue damage, blood supply, differentiation potential of bone progenitor cells, cell microenvironment, etc. Extrinsic factors include the stability of fracture fixation, fracture end spacing, inflammation, external stimulation, etc. Even social habits, such as smoking and alcohol consumption, can lead to impaired fracture healing [31].

### 3.3. Reconstruction of the Haversian System

The basic functional unit of cortical bone is the Haversian system, which is characterized by cylindrical structures [32]. Repair of the Haversian system provides the basis for mechanical reconstruction of the long bone cortex.

Bone remodeling with the primary purpose of repairing the Haversian system occurs in a special vascular structure called the bone remodeling compartment (BRC), which consists of a basic multicellular unit (BMU) [33]. The BRC is a structure with a three-dimensional pyramidal tunnel, with a cutting cone of osteoclasts, a resting zone (containing a capillary ring and supporting connective tissue in the center), and a closing cone of osteoblasts in an orderly fashion from anterior to posterior [32]. As osteoblasts deposit bone matrix to fill the cavity, vascular invagination diminishes, and the Haversian canal gradually shapes up [1,2].

During the reconstruction of the Harvesian system, the size and shape of the Haversian canal are determined by osteoclasts and osteoblasts. The cross-sectional area and circumference of a Haversian canal are positively related to the area of an osteon and the number of osteocytes within the osteon [32,34]. All of the above are the result of the osteocyte and the lacunocanalicular network (LCN) sensing and conduction system against external mechanical stress, which largely determines the size and shape of the Haversian system [35,36].

Although human and other animal bone tissues have many characteristics in common, studies have demonstrated differences between human bone and that of other mammals. For instance, humans and rabbits [37], dogs [38], sheep [39], calves [40], and monkeys [41] all exhibit secondary osteon remodeling dominated by the Haversian system, which is absent in rats and mice [15].

## 4. Animal Characteristics of Fracture Healing Models

Animal fracture studies are used to predict and explain the fracture healing process in humans, while different animal species vary in growth cycles, skeletal characteristics (including biochemical composition, bone density, mechanical strength, bone microstructure, etc.), biochemistry, and gene expression [42,43]. Fortunately, based on the consistency between the results of animal models and human clinical studies [44,45], every animal species has some specific questions that are appropriate to answer. The characteristics of commonly used experimental animals are summarized in Table 1**.**
Figure 3 illustrates the proportion of animal species that commonly used in fracture healing models in the last decade.

Bone healing mechanisms and healing rates differ between animals; small animals (mice, rats, and rabbits) are characterized by faster growth cycles, shorter bone metabolic cycles, and quicker bone healing rates than larger animals. Especially in mice and rats, intramembranous ossification with a short healing time is the main mode of repair after fracture. Stable femoral fracture healing was reported to take only 4 weeks in sexually mature rats [46]; whereas, in rabbits, dogs, and nonhuman primates, healing involves more medullary healing tissue, thus requiring a longer period [47,48,49].

The histological morphology of bone varies widely among species, with different microstructures of the periosteum, vascular distribution, and arrangement of primary and secondary osteoproteins. The details can be obtained in Brits’ review [50]. Contrary to popular belief, the skeletal structure of nonhuman primates is not similar to that of humans because the microscopic appearance of the long bones in nonhuman primates consists of avascular bone combined with irregular Haversian bone; whereas, the long bones of humans are primary vascular longitudinal bone combined with irregular or dense Haversian bone, which is similar to the skeletal structure of dogs [50]. The physical composition and fracture strength of the dog skeleton are most similar to those of the human skeleton [51], so the dog fracture model is highly applicable. Murine species do not have the Haversian system [50], so experimental studies associated with the repair of the Haversian system are not applicable.

The ideal animal fracture model should try to meet the following conditions: feasibility, reproducibility, and similarity. Feasibility refers to, first, objective feasibility: experimental animals should first be authorized ethically, and the investigator’s experimental conditions (e.g., funding, experimental period, testing conditions, etc.) should meet their needs; second, operational feasibility, the selected animals should be able to withstand the modeling measures and meet the requirements for sampling (e.g., in vivo sampling).

Reproducibility is positively related to the degree of standardization of the experimental model. The standardized ideal model can be built repeatedly without strict restrictions on the investigator, date, or study site [52,53]. Mice are difficult to standardize because of their small size, high surgical precision, and susceptibility to investigator skills.

Similarity refers to the similarity to human fractures. Specifically, the animals selected for modeling are expected to imitate the occurrence of clinical fractures as much as possible. The symptoms, signs, and examinations exhibited after modeling should be similar to those of humans. The changes in bone microstructure and bone metabolic processes in the animal model before and after modeling are comparable to those in humans as much as possible.

## 5. Classification of Fracture Healing Model Applications

### 5.1. Traumatic Bone Fracture Animal Model

Most clinical fractures are caused by trauma, such as traffic accidents, falls, and falls from height, with a large proportion of long bone diaphysis fractures [54]. At this stage, animal models for traumatic fracture healing are mainly conducted for both the femur and tibia, including open fracture and closed fracture models.

#### 5.1.1. Closed Fracture Model

A closed fracture is one in which the skin or mucosa is intact outside the fracture, and the fractured end is not in contact with the outside [55]. The closed fracture model mainly uses the principle of three-point force to simulate the force during a long bone stem fracture caused by a special fracture modeling device [56,57]. After the animal was anesthetized and fixed on the animal table, a blunt guillotine with a heavy object was dropped from a height to break the tibia or femur, and the model was successfully created. Closed fracture models are generally studied in smaller animals, such as rats and mice. Compared to mice, rats can tolerate certain surgical blows and have a larger volume of bone and serum specimens than mice, and are increasingly chosen for studies. In Einhorn’s study, a closed fracture model was made with 40 male mice; the fracture apparatus consisted of four main parts, including the overall frame, the animal fixation device, the cutting device, and a 500 g weight [58]. Since then, Einhorn’s modeling method has been refined to make it more streamlined. Subsequent studies have focused on adjusting the spring size to reduce tissue damage, adapting the device to the size of the mice, and making the device more precise in terms of impact [59,60]. In a rat model, Bonnarens and Einhorn first established a closed femur fracture model using 40 male Sprague-Dawley rats with minimal soft tissue damage from histology [61]. This model has since been refined by Simon et al., and the process is as follows: The left femur of the anesthetized rat was secured between two lower supports and an upper impactor head. A guillotine-like effect was created by dropping a rod-guided 411-g weight from a height of 20 cm onto the spring-loaded upper impactor head, creating a femoral fracture. Immediately after fracture, rats were radiographed to ensure localization of a mid-diaphyseal fracture [62,63]. Compared with open fractures, closed fracture models are easy to operate and have less impact on the surrounding soft tissues and blood flow conditions, but they cannot precisely control the fracture angle and the direction of the force line, which may have different effects on fracture healing.

#### 5.1.2. Open Fracture Model

An open fracture is one in which the skin or mucosa near the fracture is broken and the fracture end is directly or indirectly exposed [64]. The procedure for an open femur fracture is as follows: The animal is anesthetized, and an incision is made on the lateral aspect of the femur or tibia. After the skin is cut, the inner layer of the fascia and muscles are bluntly separated to fully expose the femur or tibial tuberosity; subsequently, the osteotomy is operated at different angles and positions using wire saws, electric swing saws, and other tools according to the experimental requirements to complete the fracture modeling [65,66]. For surgical procedures and outcome analysis, rabbits are most widely used in open fracture model studies because of their larger joint volume compared to rodents. Rats are also increasingly used in open fracture model studies because of the ease of husbandry, ease of purchase, relatively low cost, and tenacity of life.

The advantage of the open fracture model is that the angle and position of the fracture can be controlled, but due to the destruction of the periosteum, soft tissues, and bone marrow cavity, resulting in damage to bone cells and bone marrow stromal stem cells, blood circulation is significantly reduced, and the whole fracture healing process becomes very slow or even nonhealing.

### 5.2. Osteoporotic Fracture Model

Osteoporosis (OP) is a common systemic skeletal metabolic disease that occurs in elderly individuals and is characterized by low bone mass and destruction of bone tissue microarchitecture [67]. Osteoporotic fracture is the most common and serious complication of osteoporosis and is the most common form of fracture in elderly individuals.

#### 5.2.1. Animal Model of Osteoporosis

OP is divided into two main categories, primary and secondary, with primary OP accounting for approximately 80% of cases. Primary OP mainly includes postmenopausal and geriatric OP [68]. A common cause of secondary osteoporosis is hormone overuse. Depending on the clinical classification of osteoporosis, the establishment of animal models of osteoporosis mainly includes ovariectomy, geriatric animal models, and hormone induction [69]. The common study subjects of animal models of osteoporosis include rats, mice, rabbits, sheep, pigs, and dogs.

Postmenopausal osteoporosis model

Ovariectomy can cause a rapid decrease in estrogen levels in animals, which subsequently brings about bone loss in animals, and can well mimic the clinical features of postmenopausal osteoporosis. Ovariectomy is usually performed by making an incision on the back of the animal after anesthesia, followed by blunt separation of the superficial muscles from the peritoneum, incision of the peritoneum to expose the ovaries and fallopian tubes, ligation of the adipose tissue attached near the uterus out, clipping of the ovaries and part of the fallopian tubes, and subsequent release of the remaining adipose tissue back into the abdominal cavity for suturing.

Jilka et al. found that bone loss associated with decreased estrogen occurred soon after ovariectomy in mice, with a more pronounced decrease in cancellous bone mass [70]. The bone volume of the tibial epiphysis, vertebral body, and femoral neck will be significantly decreased 60 days after ovariectomy in rats [71]. In large animals, Chavassieux et al. found that OVX increased cortical bone porosity and surface invasion but did not affect cancellous bone in 8 ± 1-year-old sheep [72]. The rat is usually the preferred small animal model for ovariectomy because it is reproducible and simulates the decrease in estrogen levels well; it is also large enough to perform certain orthopedic surgical procedures and is better suited to assess the mechanical properties of bone [69,73].

2.Geriatric osteoporosis model

Aged animal models have been used for the study of geriatric osteoporosis. Rats over 1 year of age and sheep over 9 years of age exhibit age-related osteoporosis [74]. Mice typically survive 2–3 years with peak bone mass at 4–8 months, followed by age-related bone loss. C57BL/L and BALB/c mice can develop a geriatric osteoporosis-like bone phenotype-reduced bone mass and quality [75,76]. However, animal models of natural aging are time-consuming, and can greatly increase the cost of animal husbandry while limiting the speed of research. Accelerated aging animal models are becoming a hot topic of research in animal models of geriatric osteoporosis. The SAMP6 mouse model, a commonly used animal model for accelerated aging, shows a distinct aging phenotype at 6–8 months of age and has the potential to become a subject for the study of geriatric osteoporosis [77,78].

3.Secondary osteoporosis model

There are many causes of secondary osteoporosis, including adverse reactions to medication, endocrine disorders, eating disorders, limb wasting, kidney disease, and cancer [79]. In the current study, animal models of secondary osteoporosis mainly included hormone-induced osteoporosis and disuse osteoporosis.

Glucocorticoid induction is the most common method of modeling secondary osteoporosis. Clinically, bone density in glucocorticoid-treated patients decreases rapidly in the early phase of treatment due to enhanced bone resorption, followed by a slow decrease in bone density. Glucocorticoid-induced osteoporosis modeling is commonly used in mice, rats, rabbits, and dogs. In glucocorticoid-induced models, the age of the animal, the dose of glucocorticoid, and the duration of administration are crucial to the experimental results. Administration of weekly injections of methylprednisolone (7 mg/kg/week) to 32-week-old male Wistar rats reduced cancellous and cortical bone [80]; administration of different doses of prednisone to male rats for 90 days resulted in reduced cortical bone in the epiphysis and reduced cancellous bone formation and bone resorption [81]. In larger animals, oral administration of prednisolone (0.7 mg/kg/day) for 5 months in 6- to 7-month-old female white rabbits reduced cancellous and cortical bone [82]. Prednisolone (0.6 mg/kg/day, five times a week) injected into sheep for 7 months reduced cancellous bone volume in vertebrae and was a marker of bone formation [83]. The timing of dosing in large animal models in glucocorticoid-induced models is much closer to that in the clinical setting.

The disuse osteoporosis model is another common model of secondary osteoporosis. Disposable osteoporosis is caused by non-weight bearing, immobilization, or prolonged bed rest, and its incidence is rapidly increasing due to the increase in patients with bed-ridden disease. Disuse osteoporosis models are commonly studied in mice and are mainly modeled by tail or hind limb suspension [84]. Moriishi et al. found that 2 weeks of tail suspension experiments resulted in mild inhibition of bone formation and significantly enhanced bone resorption in C57BL/6 mice [85].

#### 5.2.2. Osteoporotic Fracture Modeling

Animal models of osteoporotic fractures are generally performed using a combination of osteoporosis and fracture models. The animals are first modeled for osteoporosis, and after determining the success of osteoporosis modeling, the corresponding fracture modeling is performed according to the purpose of the study. The most common site of osteoporotic fractures is the epiphysis. Epiphyseal fractures refer to fractures of the epiphysis at both ends of the long bones or intra-articular fractures when the fracture line spreads to the articular surface [86]. Clinically, the most common sites for epiphyseal fractures are the distal radius, proximal humerus, and proximal femur. As mentioned earlier, the rat is the most common model of osteoporosis, and it is also the most common model for epiphyseal fractures. The distal femur and proximal tibia are generally used as study sites in rat models of epiphyseal fractures. The fracture modeling procedure is usually performed by anesthetizing the animal, making an incision at the knee joint, bluntly separating the inner fascia and muscles to fully expose the distal femur or proximal tibia, and performing a vertical, transverse, or wedge osteotomy at the epiphysis to complete the fracture modeling. Alt et al. performed a 3 mm or 5 mm wedge osteotomy of the distal femoral metaphyseal region in rats 3 months after OVX to study the healing process of osteoporotic fractures [87]. The molding method of Alt has been continuously improved to make it more standardized, and the fracture gap size is adjustable to suit the needs of material-related research [88,89,90]. Nozaka et al. established a vertical osteotomy model in the proximal tibia of osteoporotic rats after OVX, with a truncated midsagittal osteotomy from the articular surface to the tibial diaphysis [91]. In large animals such as dogs and rabbits, similar to rats, epiphyseal wedge and vertical osteotomies are the most common means of modeling [47,92,93].

### 5.3. Bone Defect Model

Bone defects are an atypical type of fracture in which there is a shortage of bone due to trauma or surgery. In recent years, with the rapid development of bone tissue engineering, artificially prepared biomaterials have gradually become novel bone tissue replacements, and there are many animal bone defect models used to assess the regenerative capacity of bone replacement biomaterials [94]. Compared with the fracture model, the bone defect model is simple to operate and avoids the influence of the compensatory effect of the adjacent bone on the experiment [94]. However, the bone defect model differs greatly from the clinical reality. The animals commonly used for making bone defect models are mice, rats, rabbits, and dogs [95]. In principle, the larger the animal is, the better it matches the healing mechanism of human bone. However, the experimental cost of large animals is too high, and the number of samples is difficult to guarantee. Furthermore, the larger the animal is, the weaker its self-recovery ability, which tends to cause a long experimental cycle. The mouse model is economical and has a strong ability to resist infection and tolerate surgery, and the repair cycle is shorter and easier to manage, so the mouse model has certain advantages. The greatest advantage of rabbits over rats is that they are larger in size, more convenient to operate than rats, and within an acceptable economic range, so they are also the more commonly used animal models.

As far as the defect site is concerned, the bones used to prepare bone defect models are limb bone, cranial bone, and mandible. Cranial bones are not weight-bearing and require materials with low mechanical properties, so they are mostly used for the preparation of bone defect models with filler-type bone graft materials. Small rodents, such as mice and rats, are preferred. In rats, for example, researchers mostly use a low-speed (≤1500 rpm) sterile annular drill to create circular defects on the skull surface of rats without disturbing the dura mater. Defects ≤ 6 mm in diameter are generally drilled bilaterally at the parietal bone; defects > 6 mm in diameter are generally drilled with the sagittal suture as the center [96]. The site of bone defect preparation in the limb bone model is generally chosen as the long bone stem and bone end. The long bone trunk has a regular bone shape and is often used to prepare segmental bone defects; the long bone ends, such as the femoral condyles, the greater tuberosity of the humerus and the proximal medial tibia have cancellous bone structures and are relatively thick and blood-rich, and are often used to prepare cylindrical bone defects that do not require fixation [95].

### 5.4. Bone Nonunion

The bone nonunion model is a special kind of fracture healing model. Bone nonunion generally refers to a fracture that has been treated beyond the usual healing time and then extended again (typically 8 months after the fracture), and still fails to achieve bony healing. A variety of factors can lead to osteonecrosis, and these factors become the basis and method of osteonecrosis mapping, including bone loss at the fracture end, instability, tissue insertion, impaired blood circulation, and infection. Animal experiments and clinical studies have shown that too large a gap at the fracture end can lead to delayed healing or even nonhealing of the fracture. Claes et al. performed fracture end gap method modeling in goats and found that the larger the gap was, the slower the fracture healing [97]. Stabilization is an extremely important step in the fracture healing process, and poor stability of the fracture end tends to cause sliding of the two fracture ends, affecting tissue and vascular regeneration and osteoblast activity [98]. Volpon formed a bone defect in the middle segment of the radial shaft in adult dogs and did not use any internal or external fixation, which allowed the two broken ends to produce arbitrary rotation or sliding, destabilizing the broken ends and causing both models of hyperplastic and atrophic osteonecrosis [99]. In addition, there are other methods for modeling osteochondral nonunion, such as periostectomy and drug-induced methods. The selection of a model of osteogenesis imperfecta is based mainly on the purpose of the study.

### 5.5. Fracture Fixation Types

Fracture fixation types include intramedullary fixation and external fixation.

Intramedullary fixation

Intramedullary fixation involves the use of an intramedullary fixation pin, which is threaded into the medullary cavity of the bone stem to be fixed to control the position of that stem fracture. Studies have shown that intramedullary implants do not affect fracture healing [100,101], but inserting steel pins into the bone marrow cavity inevitably causes tissue breakdown within the cavity [102,103]. The elasticity, smoothness, diameter, and ratio of the intramedullary pin to the internal diameter of the bone marrow cavity will affect fracture healing [104,105,106]. The common method of intramedullary fixation is to insert a steel pin into the bone marrow cavity, but due to the presence of rotational torsional forces exerted on the femur by the surrounding skeletal muscles, the common intramedullary pin does not provide good stability of the femur against rotation [107]. Holstein et al. used an intramedullary pin with threads in a fracture model with 16 adult rats [105]. The surface of a normal intramedullary pin is smooth, which does not hold the bone stem axially stable and rotates the fracture part of the bone stem with the animal’s activity, which is not conducive to healing, whereas a threaded intramedullary pin provides axial stability and prevents rotation of the fracture part. Wang et al. concluded that intramedullary pins are most effective in promoting fracture healing when their modulus of elasticity is 20% to 50% of the normal femoral stiffness [106]. Intramedullary pin molding is more widely used due to the advantages of small incision and low bleeding produced in animals. In addition, the materials used for intramedullary fixation pins have become more diverse in recent years as research has progressed, including new alloy materials and biodegradable materials, etc. [108,109,110].

2.External fixation

External fixation refers to the technique of inserting steel pins into the proximal and distal ends of the diaphysis and connecting the pins with metal or high-strength nonmetallic rods and connecting devices outside the body to achieve the effect of fracture treatment, correction of diaphysis and joint deformity, and limb lengthening through fixation, compression, and traction. External fixation molds provide greater resistance to rotational torque and do not cause significant damage to the tissue within the bone marrow cavity [111,112]. The disadvantage is that the external fixation molding process is more complicated, the molding time is longer, the bleeding is high, and it is likely to lead to the death of the animal. In addition, external fixation molds are similar to putting a pair of shackles on the animal, and the more weight it has, the more it affects the animal [57]. The steel pin used for external fixation directly connects the bone tissue to the outside world, which can easily cause infection inside the bone tissue [113].

**Table 1 bioengineering-10-00201-t001:** The characteristics of commonly used experimental animals.

Animal Species	Sexual Maturity Time	Epiphyseal Plate	Haversian System *	Advantages	Disadvantages	Applicable Models
Mouse	6 weeks	Existence For a Lifetime	None	Low price, easy feeding, short growth cycle. Detailed genome map. Plenty of monoclonal antibodies are available.	Small size, high surgical technique requirements, difficult to standardize, and poor reproducibility. Hard to evaluate biomechanical indicators and serum indicators. Cannot be implied in Haversian system research.	Model of delayed healing and nonunion [114]. Closed and open fractures [58,115]. Epiphyseal fractures [116]. Bone defects [117].
Rat	8 weeks	Thinning with age	Rare	Low price, easy feeding. Strong vitality, withstand surgery. Easy to evaluate biomechanical indicators and serum indicators. Thymus-free rats can be used for bone transplantation studies.	Their bone differs greatly from human bones in biochemical composition, density, and mechanical capacity. Hard to implied in Haversian system research.	Closed and open fracture model [61,118]. Epiphyseal fractures [83]. Bone defects [119].
Rabbit	4–6 months	Closure after sexual maturity	Existence after sexual maturity	Gentle temperament, easy feeding. Low surgical technique requirements. Easy to do biomechanical analysis. Easy to draw blood. Haversian system similar to that of humans [120]. Rapid skeletal transformation [120].	Biomechanical aspects differ greatly from that of human bones.	Femoral head necrosis model [121]. Fatigue fractures model [122]. Open fracture model [123]. Epiphyseal fractures [124]. Bone defects [125].
Dog	8–10 months	Closure after sexual maturity	Existence	Sexually mature dogs’ bones are largely similar to those of humans in physical properties, physiological characteristics, and fracture strength [51].	Biomechanical reconstruction is different from that of humans.	Model of delayed healing and nonunion [99]. Open fractures [126]. Epiphyseal fractures [127]. Bone defects [128].
Sheep and goat	10–12 months	Closure after sexual maturity	Existence	Docile, easy to handle, relatively inexpensive, spontaneously ovulate. The hormone profiles are similar to that of women [129].	Lack of natural menopause, that normal estrus cycles are restricted to fall and winter. The gastrointestinal system differs greatly from that of humans [129].	Open fracture model [130]. Epiphyseal fractures [131].
Pig	4–6 months	Closure after sexual maturity	Existence	Easy to acquire specimens (i.e., blood, urine, bone). Bone density and fracture strength are similar to human.	High price, not easy to feed. Biomechanical reconstruction is different from that of humans.	Open fracture model [132].
Nonhuman primates	3–5 years	Closure after sexual maturity	Existence	Biomechanical characteristics closest to those of humans. The fracture fixation and biomechanical reconstruction are similar to human. Moderate body size and low surgical technique requirements.	High price, short growth cycle, not easy to feed. Their sources are very limited due to the strict ethical and legal protection of animals.	Various fracture healing models (particularly suitable for studies related to mechanics, osteoconductivity) [133].

* The information comes from Hillier et al. [134] and O’Loughlin et al. [135].

## 6. Summary

Animal fracture models are designed to study the mechanisms of bone tissue repair during fracture healing. Fracture healing is a very complex process, and no all-inclusive model can accurately mimic the underlying osteoporosis or fracture pattern in humans. This review found that rodents and rabbits are more popular in the choice of animal species because of economic factors and suitability. The choice of animal models for fracture healing varies depending on the purpose of the study, but long bone fractures remain the most dominant fracture model. Furthermore, there are still some challenges in animal fracture models: (1) existing studies mainly focus on long bone fractures, and the understanding of special site fractures (including epiphyseal fractures, joint fractures, etc.) is not yet comprehensive, and there is no standardized surgical protocol; (2) osteoporotic fractures have not yet been able to fully simulate the clinical spontaneous fracture process in humans, especially vertebral compression fractures, while no effective small animal model has been found. These problems still need to be solved by further research, and we believe that with the continuous development of science, new animal fracture models will emerge.

## Figures and Tables

**Figure 1 bioengineering-10-00201-f001:**
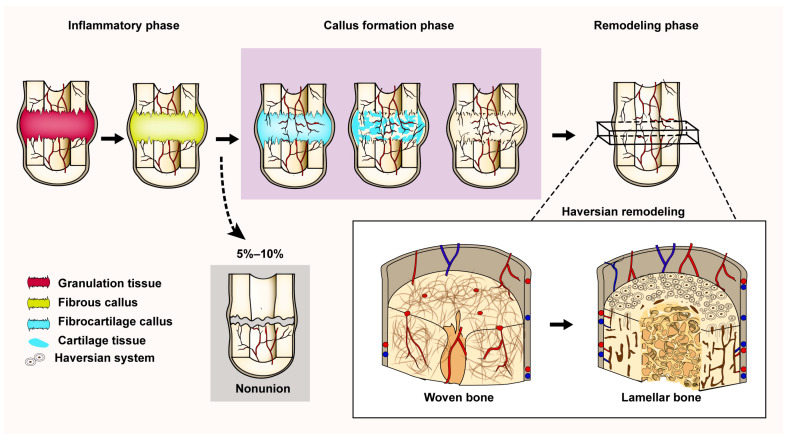
The process of endochondral ossification. Fracture repair begins with the inflammatory phase, where the hematoma at broken end of fracture is replaced first by granulation tissue and subsequently by fibrous callus. During the proliferative phase, proliferation of blood vessels and osteoprogenitor cells occurs, leading to the formation of fibrocartilage tissue. Next, the fibrocartilage callus is gradually replaced by woven bone and the excess callus is resorbed by osteoclasts. At last, the transformation of woven bone to lamellar bone is accomplished by remodeling (reconstruction of the Haversian system).

**Figure 2 bioengineering-10-00201-f002:**
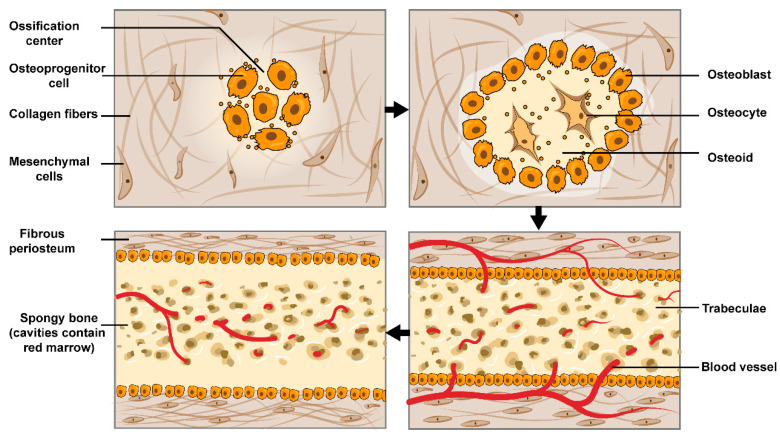
The process of intramembranous ossification. A small group of adjacent mesenchymal cells form a small cluster of cells and undergo morphological changes to form osteogenic progenitor cells. These cells gradually differentiate into osteoblasts and secrete extracellular matrix (osteoid) and mineralize, resulting in spicules formation. As the secretion of osteoid increases, the spicules increase in size and fuse with each other, leading to the formation of trabeculae. As growth continues, the trabeculae interconnect and then form spongy bone. Finally, the mesenchyme at the periphery of the trabeculae forms the periosteum.

**Figure 3 bioengineering-10-00201-f003:**
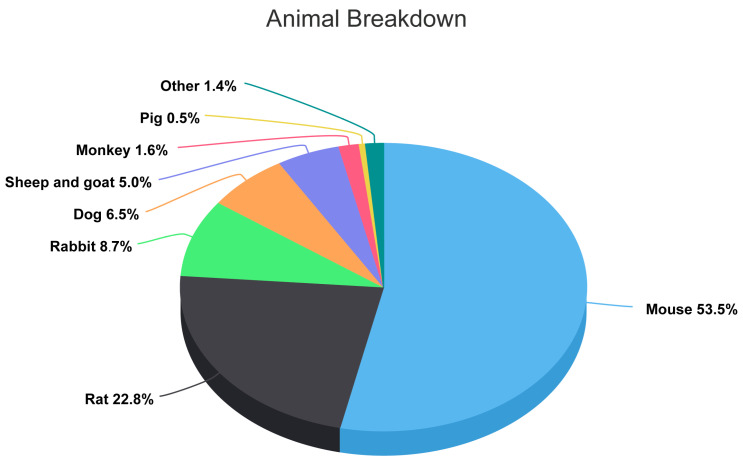
The pie chart represents the proportion of animal species commonly used in preclinical fracture studies over the past decade.

## Data Availability

Original data are available on personal request from the corresponding author.

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
