# Peer review of "Advances in Animal Models for Studying Bone Fracture Healing"

_bioengineering, 2023, doi:10.3390/bioengineering10020201_

Round 1

Reviewer 1 Report

Gao et al. submitted a manuscript for a review paper concerning animal models for bone fracture healing study. The authors aimed to gather current status of fracture healing researches and to provide a guideline for basic experimental fracture modeling.

However, overall detailed information lacks and there are many typos and misused terms throughout manuscript. Figure1 is not convincing. Authors should provide comprehensive intramembranous and endochondral ossification process. Table 1 is difficult to follow and details are largely lacking. Especially, there are several fracture models using mouse to study bone healing. In addition, how the authors decided with “Advantages” and “disadvantages” is not clear.

Author Response

Answer:

    Thank you very much for your comments! We have adjusted the language for publication and corrected the use of terminology. In the meantime, we rewrote the section on "Fracture healing mechanisms", and modified Figure 1 to describe the fracture healing process in a more detailed way, and simplified Table 1. In addition, the “Advantages” and “disadvantages” of the animals are summed up from general experience just like Padhraig’s review (http://dx.doi.org/10.2106/JBJS.G.01585), due to there is no standardized criterion for assessing this.

Reviewer 2 Report

The article is sound, and the references search, likewise. The title might be misleading since the reader thinks it is about all kinds of fractures, and it is not. The article mainly deals with osteoporotic fractures. Skull fractures are not taken into account and those are the main ones due to violence or accidents. This issue has to be clarified. Then bone remodeling is much more complex than stated, depending on both intrinsic and extrinsic factors. A correct evaluation of healing should be done using a holistic approach with histology, and imaging, among others. The authors should aid that. Besides, nowadays, PSTI post-traumatic time interval is a significant issue in forensic sciences. there is a need to know when the injury was done. This issue is either approached in this article, and it should have been discussed. See, ie, Hans de Boer articles. Then the table is too long, and the reader gets lost, would it be possible to shorten it? I did minor suggestions in the manuscript using pdf comments. Falls from height are a type of falls.

Author Response

Answer: 

Thank you very much for your comments! This review mainly summarizes the most common fracture animal experimental models. Simple skull fracture model is relatively rare in experimental animals, and skull fracture is more common in closed craniocerebral injury model in animal models, which is not suitable for discussion in this review. PSTI post-traumatic time interval is a significant issue in forensic sciences, but this is not generally described in animal models. In addition, you mentioned in your comments that bone remodeling is much more complex than we stated, so we rewrote this part (page 4, line 21 to page 7, line 9) In this review, we hope to present the modelling methods and characteristics of different fracture animal models to help researchers in animal selection, and we have simplified Table 1.

Reviewer 3 Report

The Review „ Advances in animal models for studying bone fracture healing” by Hui Gao and co-workers investigates the role of animal models in the course of the last 20 years. Although the review focusses on a defined search pattern, many aspects are lacking depth and need revision as I will discuss now in more detail

1.       Introduction

Because in the next parts the Haversian channel is so important, some information of the role and the importance for fracture repair would be helpful.

2.       Search strategy

No objections, this is a valid strategy

3.       Mechanism of fracture healing

Why did the authors think that the healing mode is dependent on the Haversian System?

The explanation of the morphological process is very superficial with just naming some growth factors but without naming their role

4.       Animal characteristics of fracture healing models

Table 1 is in my point of view not correct. Especially for mice and rats, they are readily used in several other methods, especially closed and open fractures, as you write it later on. And again, the stress on the Haversan system is not clear, why this is an issue. Finally the layout of this table is not very clear and needs improvement.

Is Haversian and Haverds system the same or is this a mistake?

5.1. Classification of fracture healing model applications  

5.1.1. Traumatic bone fracture animal model

Here the numbering went completely off, please check and correct your formatting.

Open fracture model

After the traumatic fracture model, I would start with the closed fracture model, since it is still somehow traumatic as well. Open fracture model and bone defects are similar on answering similar scientific questions.

The word osteotomy, which is the core procedure of this model in uttered much later and has to be used in the right context in this paragraph

Additionally, and this goes through the whole review, there is no word about stabilization. This is applicable for every model you mentioned with exception of “bone defects where a stable defect is set.

Also there is nothing said about healing differences, which model in which animal need more healing time. What is an enchondral healing model which is an intramembranous? How these models can be modified to induce one or the other healing mode!

Options like non-union models are not addressed here.

Closed fracture model

Here you cite the Einhorn Paper which used mice for an closed fracture model. And this does not appear on table 1. No word to stabilization, healing time, monitor healing (imaging)

5.2. Osteoporotic fracture model

This is not a separate fracture model but a special group of sample animals.

5.3. Bone defect model

“However, the bone defect model differs greatly from the clinical reality, and the reproducibility is low.” – The second path is not true. Especially since the reproducibility is so high, this method is done. But you are right this model is not suited for answering clinical questions. But for addressing mechanistic problems in bone biology it is a well suited mechanism because other influences like biomechanics, inflammation etc. if these are not topic of the research, do not interfere with this model. Here again the lack of addressing stabilization, which are a big issue in fracture animal models is obvious.

6. Summary

“Whether internal fixation methods such as intramedullary pins can cause changes in fracture-related factors and affect the fracture healing process. These problems still need to be solved by further research, and we believe that with the continuous development of science, new animal fracture models will emerge.“ – This is not sufficient, since internal as well as external fixation has a big impact on healing time and outcome and vary greatly from model to model!

With all these remarks, I think this review is not suited for publication and need heavy reworking especially with formatting and addressing the topics I have mentioned. Two main things are most crucial which are the role of the Haversian channels in the fracture healing models and the role of proper stabilization, especially with the focus on the need for clinical translation

Author Response

3.The Review „ Advances in animal models for studying bone fracture healing” by Hui Gao and co-workers investigates the role of animal models in the course of the last 20 years. Although the review focusses on a defined search pattern, many aspects are lacking depth and need revision as I will discuss now in more detail

Answer:

Thanks very much for your comments, which are very helpful to promote our paper.

  1. Introduction

Because in the next parts the Haversian channel is so important, some information of the role and the importance for fracture repair would be helpful.

We added information on the Haversian system, the role and importance of fracture repair in the introduction. Thank you for your advice.

2.Search strategy

No objections, this is a valid strategy

  1. Mechanism of fracture healing

Why did the authors think that the healing mode is dependent on the Haversian System?

The explanation of the morphological process is very superficial with just naming some growth factors but without naming their role.

Answer: 

We rewrote the section on "Fracture healing mechanisms", and modified Figure 1 to describe the fracture healing process in a more detailed way, and simplified Table 1

  1. Animal characteristics of fracture healing models

Table 1 is in my point of view not correct. Especially for mice and rats, they are readily used in several other methods, especially closed and open fractures, as you write it later on. And again, the stress on the Haversan system is not clear, why this is an issue. Finally the layout of this table is not very clear and needs improvement.

Answer: 

We rewrote the section on "Fracture healing mechanisms"(page 4, line 21 to page 7, line 9), and introduced “Three key elements in fracture healing”, “the process of indirect fracture healing”, and “Reconstruction of the Haversian system”. Then we modified Figure 1 to describe the fracture healing process in a more detailed way, and simplified Table 1

Is Haversian and Haverds system the same or is this a mistake?

Answer: 

I am very sorry that this is our oversight. The correct expression is Haversian and we have modified it.

5.1. Classification of fracture healing model applications  

5.1.1. Traumatic bone fracture animal model

Here the numbering went completely off, please check and correct your formatting.

Answer: 

We have checked and corrected the numbering.

Open fracture model

After the traumatic fracture model, I would start with the closed fracture model, since it is still somehow traumatic as well. Open fracture model and bone defects are similar on answering similar scientific questions.

Answer: 

We have made adjustments according to your suggestion.

The word osteotomy, which is the core procedure of this model in uttered much later and has to be used in the right context in this paragraph

Answer: 

We have made adjustments according to your suggestion(page 9, Chapter 5.1.2).

Additionally, and this goes through the whole review, there is no word about stabilization. This is applicable for every model you mentioned with exception of “bone defects where a stable defect is set.

Answer: 

We have added information about the method of fracture fixation.

Also there is nothing said about healing differences, which model in which animal need more healing time. What is an enchondral healing model which is an intramembranous? How these models can be modified to induce one or the other healing mode!

Options like non-union models are not addressed here.

Answer: 

We have added to the introduction of nonunion (page 2, line 18-20).

Closed fracture model

Here you cite the Einhorn Paper which used mice for an closed fracture model. And this does not appear on table 1. No word to stabilization, healing time, monitor healing (imaging)

Answer: 

We have added information about the method of fracture fixation.

5.2. Osteoporotic fracture model

This is not a separate fracture model but a special group of sample animals.

Answer: 

The bone healing mechanism of osteoporosis animals is different from that of ordinary animals. Although it is a special animal model, it is a common problem in clinical practice. Therefore, we summarized in this paper.

5.3. Bone defect model

“However, the bone defect model differs greatly from the clinical reality, and the reproducibility is low.” – The second path is not true. Especially since the reproducibility is so high, this method is done. But you are right this model is not suited for answering clinical questions. But for addressing mechanistic problems in bone biology it is a well suited mechanism because other influences like biomechanics, inflammation etc. if these are not topic of the research, do not interfere with this model. Here again the lack of addressing stabilization, which are a big issue in fracture animal models is obvious.

Answer: 

We couldn't agree with you more. Here we refer to the phalangeal defect model as having a low analog for clinical problems. Perhaps because of the lack of English expression, we can't express our views clearly here. In addition, we have added to the article fixed mode related content ((page 16, Chapter 5.5).

  1. Summary

“Whether internal fixation methods such as intramedullary pins can cause changes in fracture-related factors and affect the fracture healing process. These problems still need to be solved by further research, and we believe that with the continuous development of science, new animal fracture models will emerge.“ – This is not sufficient, since internal as well as external fixation has a big impact on healing time and outcome and vary greatly from model to model!

Answer: 

We did not consider this issue thoroughly enough and have removed the relevant content.

With all these remarks, I think this review is not suited for publication and need heavy reworking especially with formatting and addressing the topics I have mentioned. Two main things are most crucial which are the role of the Haversian channels in the fracture healing models and the role of proper stabilization, especially with the focus on the need for clinical translation.

Answer: 

we added information on the Haversian system, the role and importance of fracture repair in the introduction.We have added information about the method of fracture fixation.

Round 2

Reviewer 1 Report

The authors have improved the manuscript. However, there are still several points that have to be addressed clearly.

Figure 1:

·      Authors should revise the stages and images according to known concepts (e.g. Inflammatory, endochondral/intramembranous ossification (or “callus formation”), remodeling phase).

·      The image for intramembranous ossification can be misleading, because the cartoon looks just similar to endochondral ossification. Intramembranous ossification occurs usually at the distal and proximal ends of the fracture, or bone defects (e.g. tibial or calvarial defects).

·      And authors placed “progenitor cells” only in the intramembranous ossification image, which is not acceptable. Skeletal stem/progenitor cells are recruited in the injury site and they will participate into both endochondral and intramembranous ossification.

·      The plane for “haversian remodeling”(that you marked in “Remodeling phase” bone image) and images with “Woven bone” and “Lamellar bone” need to be revised. “Woven bone” looks like a cortical bone, not a trabecular one. “Lamellar bone” is also hard to understand from the image.  

·      Figure legend with concise explanation should be inserted.

5.5. Fracture fixation types

·      External fixation: “The disadvantage is that the external fixation molding process is more complicated, the molding time is longer, the bleeding is high and it is likely to lead to the death of the animal. In addition, external fixation molds are similar to putting a pair of shackles on the animal, and the more weight it has, the more it affects the animal. The steel pin used for external fixation directly connects the bone tissue to the outside world, which can easily cause infection inside the bone tissue.” Can authors provide any references for these sentences?

Table 1:

·      Authors mentioned that the information of the table 1 is from Hillier et al. [109]. However, Hillier et al. paper is a review paper concerning differences between human bone and other animal bone based on histological methods. Authors might have a mistake to add a right reference, since they replied to me with other citation (http://dx.doi.org/10.2106/JBJS.G.01585). Please add this one, too.  

·      Within the table, please cite the research papers that demonstrated indicated fracture models in specific animal species (e.g. add a column for references in the end of the table).

Author Response

Thank you very much for your comments!

  1. We modified the figures of intramembranous ossification and endochondral ossification with figure legends. In order to make the schematic diagram easier to understand, we have added details of the “Remodeling phase” (including “Haversian system”, “Woven bone” and “Lamellar bone”) in Figure 1 (page 5). We also newly drawn the image of endochondral ossification according to known concepts (Figure 2) (page 6).
  2. We have added references in the corresponding places in "5.5 Fracture fixtion types".

  3. We added the literature citation of O'Loughlin’s paper (http://dx.doi.org/10.2106/JBJS.G.01585) in the table 1.

  4. We cited the research papers that demonstrated indicated fracture models in specific animal species at the last column of the table.

Reviewer 3 Report

The authors have made substatial corrections on their review. I have to ask for two additions nevertheless.

1. 5.5. Intermedulary stabilization: RI Systems in Switzerland provides with mouse screws which are implantats that tackle the problem of rotatation stabilization. Especially for small rodents. Also there are publications which investigate the role of stabilization in imaging (MRT and CT). That's why the authors should mention different implantat materials besides steel

2. Table 1 The healing mode (Intramembranous/enchondral ossification) is not dependent on the animal model but on the distance of fractured bone. Thus you can not exclusively say that one mode is used in one model. Our group use enchondral ossification in murine model all the time.

Please comment on these correction and then I think the review is fit for publication

Author Response

Thank you very much for your comments!

1.We have added additional material intramedullary pins in paragraph 5.5.

2. We agree your comment “the healing mode (Intramembranous/enchondral ossification) is not dependent on the animal model but on the distance of fractured bone”, so we deleted the conflicting content in Table 1 and made additions in the manuscript (page 4, line 28-page 5, line 2).

Round 3

Reviewer 1 Report

The authors answered all my concerns and improved well their manuscript.

However, there are few things to be corrected. 

Figure1: “Proliferative phase” can be replaced by “Callus formation phase”, since “Proliferation” occurs in different cell types already before “fibrous callus” formation.

Figure number “1” is repeated for second figure in the figure legend.

Author Response

Thank you very much for your patient work!

We replaced “Proliferative phase” with “Callus formation phase” in figure 1 and corrected the figure legend number in figure 2.